# Prevalence and Factors Associated with High Concentration of Prostate-Specific Antigen: ELSIA Study

**DOI:** 10.3390/biology9100329

**Published:** 2020-10-09

**Authors:** Lucas Lima Galvão, Sheilla Tribess, Tamara Guimarães Silva, Cremilda Garcia Santa Rosa, Cristian Gomes Pereira, Rizia Rocha Silva, Jeffer Eidi Sasaki, Jair Sindra Virtuoso Junior, Claudio Andre Barbosa de Lira, Douglas Assis Teles Santos

**Affiliations:** 1Departamento de Ciências do Esporte, Universidade Federal do Triângulo Mineiro, Uberaba, Minas Gerais 38025-180, Brazil; d201910782@uftm.edu.br (L.L.G.); sheilla.tribess@uftm.edu.br (S.T.); d201910785@uftm.edu.br (R.R.S.); jeffer.sasaki@uftm.edu.br (J.E.S.); jair.junior@uftm.edu.br (J.S.V.J.); 2Laboratório Municipal Nova Filosofia, Rede LACEN-BA, Teixeira de Freitas, Bahia 45995-000, Brazil; tamistgs@gmail.com (T.G.S.); cremilda.srosa@kroton.com.br (C.G.S.R.); 3Colegiado de Educação Física, Universidade do Estado da Bahia, Teixeira de Freitas, Bahia 41150-000, Brazil; cristiangpeducacaofisica@gmail.com; 4Faculdade de Educação Física e Dança, Universidade Federal de Goiás, Goiânia, Goiás 74690-900, Brazil

**Keywords:** elderly, health behavior, ageing, sedentary behavior

## Abstract

**Simple Summary:**

Screening for prostate cancer is critical to increasing men’s longevity, and prostate-specific antigen is the primary method of screening for this cancer. Assessing the prevalence and factors associated with high concentrations of this antigen is essential and this was the objective of our study. We found associations between high concentrations of prostate-specific antigen with years of study, race/ethnicity and family arrangement, health perception, systolic blood pressure, diastolic blood pressure, metabolic diseases, alcohol consumption and sedentary behavior. These findings may guide public health policies in order to create guidelines that raise awareness to reduce risk behaviors that increase the concentrations of prostate-specific antigen.

**Abstract:**

Background: Prostate cancer (PC) is the second most common cancer among men, behind only non-melanoma skin cancer, and the main method of screening for PC is the prostate-specific antigen (PSA). To analyze the prevalence and the factors associated with high concentration of PSA in the elderly is essential to understand this outcome, and building strategies to decrease their rates of morbidity and mortality. Methods: We performed a cross-sectional study with 96 elderly men. A high level of PSA was defined by >4.0 ng/mL. In order to identify sociodemographic, health, functional and behavioral variables, which may be associated with high levels of PSA, we carried out a multivariate analysis using Poisson regression. Results: The prevalence of high levels of PSA was 21.9% (*n* = 21). High levels of PSA was associated with years of study, race/ethnicity and family arrangement, health perception, systolic blood pressure, diastolic blood pressure, metabolic diseases, alcohol consumption and sedentary behavior. Conclusions: The study found a high prevalence of high PSA concentrations in the elderly and several aspects are associated, which can be a worrying factor for their health, since PSA is an important marker of prostate cancer.

## 1. Introduction

Prostate cancer (PC) is the fifth most common cancer in the world, being the second most frequent among men, behind only non-melanoma skin cancer [1,2,3]. Its incidence has been higher in developed countries compared to less developed countries [1,2], being more common in men over 50 with the risk increasing with advancing age, with approximately 75% of cases in elderly people over 80 years [1].

In Brazil, an estimated 65,840 new cases of PC in the 2020–2022 triennium are estimated, corresponding to a risk of 6295 cases per 100,000 men [4], with a mortality rate ranging from 55.4 to 114.3 deaths per 100 thousand men, in the period from 1980 to 2012 [5].

Studies have shown its etiology related to dietary factors (high consumption of fats and fibers, low consumption of fruits and vegetables) [6] and hormonal changes (androgens and insulin-like growth factor 1 [IGF-1]) [1]. It has been suggested that the hereditary component of PC is more important than in other cancers, in part believing that 40% of cases are linked to hereditary factors [1,6,7], with morphological (prostatic volume, height, obesity) [7,8,9,10] and behavioral factors (low level of physical activity and long time in sedentary behavior (SB)) also being significant [11,12,13]. However, smoking and drinking have not yet been strongly associated with the onset of cancer [1,11].

The main methods of screening for PC are digital rectal examination, sextant prostate biopsy and prostate-specific antigen (PSA) [6,14]. PSA is a glycoprotein produced by prostatic epithelial cells, secreted in the seminal plasma, where it is present in high concentrations, and in lower concentrations in the serum of men without prostatic changes. It is used to screen for PC and monitor the results of treatment [15]. Its levels can be influenced by prostatic changes and non-malignant changes (age, use of drugs, inflammation of the prostate, race/ethnicity, weight, body mass index (BMI), obesity and diabetes) [4,13,15].

Due to the importance of PSA in the detection of PC, it is essential to verify factors associated with its high concentrations, in order to decrease invasive procedures, and decrease morbidity and mortality rates and cancer problems in the elderly male population, especially in the Brazilian population where studies are scarce. Thus, the aim of this study was to analyze the prevalence of high levels of PSA and its association with sociodemographic variables, health aspects, functional and behavioral aspects in the elderly.

## 2. Materials and Methods

### 2.1. Study Design

This study is part of the Longitudinal Health Study for the Elderly of Alcobaça-Bahia (ELSIA). This investigation is characterized as an observational study of cross-sectional and analytical type, using exploratory methods such as surveys and blood collection, carried out in Alcobaça city, Brazil. The study protocol and procedures are in accordance with the Declaration of Helsinki and were previously approved by the Human Research Ethics Committee of the Federal University of Triângulo Mineiro on 27 February 2015 (ethics code: 966.983).

### 2.2. Participants

The population of the municipality was 21,319 inhabitants, with 2,047 people aged 60 or over, of whom 1024 represented the total of elderly residents in the urban area of the municipality [16].

The initial sample of the present study consisted of 743 elderly people registered in the Family Health Strategy, in Alcobaça city. During data collection, 54 elderlies refused to participate, 58 were excluded for not meeting the inclusion criteria and 158 elderlies were not located after three attempts. Of a total of 473 subjects ≥ 60 years old included in the study, 178 were male, of whom 96 (*n*) presented information about the outcome variable (PSA) composing our final sample.

The exclusion criteria were: having severe cognitive impairment in the Mini Mental State Examination (MMSE), adapted for the Brazilian population [17], severe difficulty in visual and hearing acuity, use of wheelchairs, having severe sequelae of stroke with localized loss of strength or having a terminal illness.

For the home visit, the researchers used data provided by the Municipal Health Department of Alcobaça as a reference [18]. The contact was made with the elderly through home visits, informing the objectives and requesting their participation in the research on a voluntary basis.

### 2.3. Sociodemographic Variables

The sociodemographic variables investigated were: age; years of study (literate and illiterate), monthly family income, race/ethnicity (white, black, brown) and family arrangement (alone or accompanied).

### 2.4. Health Aspects

The variables of health aspects were composed of systolic blood pressure (SBP) and diastolic blood pressure (DBP), obtained in an interval of 5 min by means of a digital sphygmomanometer Omron–HEM 7113 (Omron, Matsusaka, JPN/Mie, Japan. Perception of health (positive or negative) and metabolic diseases (yes or no) were assessed using self-report.

### 2.5. Functional and Behavioral Aspects

Alcohol consumption was self-reported by asking questions about the consumption of alcoholic beverages frequently, at least once a week, being dichotomized as yes or no.

The assessment of the level of moderate and vigorous physical activity (MVPA) and SB was carried out according to the International Physical Activity Questionnaire (IPAQ) validated for the elderly Brazilian population [19]. The MVPA was determined from the four domains of physical activity, leisure, work, transportation and household. The SB was determined by the time spent sitting, assessed from the questions of sitting time on a usual day of the week and a usual weekend day. The total time spent sitting, minutes/day, was determined from the weighted average of the time spent sitting on a weekday and a weekend day: [(sitting time on a weekday × 5 + sitting time on a weekend day × 2) /7].

Flexibility was assessed using the sit and reach test, using the senior fitness test battery [20], recording the distance (cm) to the toes as zero point (0), the distance before it was adopted as negative (−) and positive (+) for distances beyond the feet.

The calf circumference was assessed using a flexible and inelastic measuring tape (Lange-TBW, Cambridge, MA, USA), 2 m long, graduated in centimeters and subdivided in millimeters, evaluating the area with the largest protuberance in the region [21].

Muscle mass was obtained using the equation for quantification, represented by Total Muscle Mass (TMM), an equation validated by Rech et al. [22] for the elderly Brazilian population. The proposed equation makes use of body mass (kg), height (m), gender, age and race/ethnicity. In addition, for the variable gender: 0 = women and 1 = men, and for race/ethnicity the values 0 = white (white, mestizo and indigenous), 1.2 = Asian and 1.4 = Afro-descendant (black and mulatto). For the sake of standardization, the Afro-descendant was characterized by black.
TMM (kg) = (0.22 × body mass) + (7.8 × height) − (0.098 × age) + (6.6 × gender) + (race/ethnicity − 3.3).

### 2.6. Prostate-Specific Antigen

Blood samples were collected in the morning using vacuum blood collection tubes with 10 mL plastic clot activator with a diameter of 16 × 100 mm, and the elderly were instructed to remain fasting for eight hours, not to have sexual intercourse and to avoid horse displacement, motorcycle and bicycle use for 72 h before collection. After venipuncture, the blood was stored, to be subsequently centrifuged (procedure carried out in a balanced way, checking the calibration and matching the tubes according to their physical characteristics) in pairs for 10 min at 3000 rpm, using an Inbrás ALB 18 VT Serological centrifuge (Inbrás, Ribeirão Preto, BRAZ/SP, Brazil), then the serum was transferred with the aid of an automatic pipette and variable volume to a tube properly labeled with a minimum of 2.0 mL of serum, and stored in an ice box and then transferred to the laboratory that performed the sample processing, following the Standard Operating Procedure (SOP) [23] and ARCHITECT i1000SR immunoassay analyzer (Abbott, Green Oaks, USA/IL, United States of America). A PSA level equal to or greater than 4 ng/mL was considered high [13,15,24].

### 2.7. Statistical Analysis

To make the database, the software Epidata, version 3.1b (EpiData Association, Odense, DK/Fiónia, Denmark), was used, and for the analyses, the statistical software SPSS 23.0 (Statistical Package for the Social Sciences, IBM, Chicago, USA/IL, United States of America) was used. The normality of the data was verified by the Kolmogorov–Smirnov test for all analyzed variables.

To identify the factors associated with PSA, crude and multivariate analysis was performed with estimates of the prevalence ratios through Poisson regression. The crude models were constructed containing each of the independent variables and the response variable (high PSA level > 4.0 ng/mL). The variables for which *p* < 0.20 (Wald test) values were obtained were candidates for multiple models (hierarchical). To calculate the adjusted prevalence ratios (PR), a significance level of 5% and a confidence interval (CI) of 95% were considered.

Through the established strategy of associations between the dimensions studied, an explanatory model was elaborated that used Poisson regression, introducing the variables in the form of blocks and controlled by the variables age and income.

In block 1, are the sociodemographic variables: years of study, race/ethnicity and family arrangement with them, in block 2 there are health aspects: health percetion, SBP and DBP and metabolic diseases, and in block 3, the variables of functional aspects (flexibility of lower limbs and calf circumference) and behavioral aspects (consumption of alcohol, physical activity in transport, SB and TMM).

Descriptive statistics procedures were used to identify the sample with frequency distribution, calculation of central tendency and dispersion measures (range of variation, standard deviation (SD)). The Chi-square test was used to verify the distribution of PSA in relation to SB. The significance level of 5% was adopted.

## 3. Results

The characteristics of the elderly are shown in Table 1. The prevalence of high PSA levels was 21.9% (*n* = 21).

Table 2 shows the crude and multivariate Prevalence Ratio (PR) for the independent variables (sociodemographic, health, and functional and behavioral aspects) in relation to PSA in the elderly, controlled by the variables age (in continuous years) and monthly family income (amount of minimum wages, in Brazil, in 2015, equivalent to R$788.00).

In the crude analysis, the PSA was significantly associated with years of study, race/ethnicity, family arrangement, perception health, SBP and DBP, metabolic diseases, alcohol consumption, physical activity in transport, lower limb flexibility, circumference of calf, TMM and SB.

The multivariate analysis, controlled by block 1—sociodemographic variables, block 2—health variables and block 3—functional and behavioral aspects respectively, when adjusted for age and monthly family income variables, remained significantly associated with the prevalence of PSA. The variables years of study (PR = 0.23; 95% CI = 0.10–0.95), race/ethnicity (PR = 0.26; 95% CI = 0.08–0.82) and family arrangement (PR = 0.17; 95% CI = 0.07–0.38) are considered a protection to high levels of PSA.

Health perception (PR = 2.45; 95% CI = 1.03–5.82), SBP (PR = 1.02; 95% CI = 1.01–1.04) and DBP (PR = 1.03; 95% CI = 1.00–1.06), metabolic diseases (PR = 2.35; 95% CI = 1.02–5.40), alcohol consumption (PR = 4.06; 95% CI = 1.77–9.31) and SB (PR = 1.00; 95% CI = 1.00–1.01) are considered risk factors. The variables of physical activity in transport, calf circumference and lower limb flexibility were not associated in the adjusted analysis.

## 4. Discussion

The aim of the study was to analyze the prevalence of high levels of PSA and its association with sociodemographic variables, health aspects, functional and behavioral aspects in the elderly. The main results of the study were: (i) high prevalence of high PSA levels and (ii) years of study, race/ethnicity, family arrangement, health perception, SBP and DBP, metabolic diseases, alcohol consumption and SB were factors associated with high PSA levels.

Regarding the prevalence of high PSA levels, Van Hemelrijck et al. [25], in their research aimed at assessing the nutritional status of the non-institutionalized population in the USA, evaluating 1312 men and of these 1.6% (*n* = 21), presented high levels of PSA, however, the cutoff point adopted was ≥10 ng/mL. While, Loprinzi and Kohli [13], using the cutoff ≥4.0 ng/mL, found a prevalence of 6.2% in 1672 non-institutionalized men, the elderly, black non-Hispanic and Mexican-Americans. The prevalence found in our study (21.9%) may have been higher, due to the cutoff point adopted and the greater coverage of the population.

Participants were dichotomized into literate and non-literate. Literate participants had a lower prevalence ratio of high PSA levels compared to non-literate ones, a result also obtained by Loprinzi et al. [26] and Loprinzi and Kohli [15]. This fact can be caused by the impact of low years of study being associated with worse lifestyle habits, such as low level of physical activity [27] and worse quality of life scores in different domains [28]. Our results are still in line with Loprinzi and Kohli [13] for demonstrating that men living with a partner have lower concentrations of PSA. In our study, living together was shown to be a protective factor for high PSA levels (PR = 0.17; 95% CI = 0.07–0.38).

The literature shows that black (African-American) individuals are more likely to have high PSA levels and develop PC than their white counterparts [9,13,24], including in Brazil [4]. Giovannucci et al. [7] conducted a follow-up study with men and demonstrated that African-American men had a higher incidence of PC. Romero et al. [29] evaluated 1,544 men and observed a 4.0% higher prevalence of high PSA levels (≥4 ng/mL) in African and black descendants compared to whites and non-African descendants. In our study, there was no association between high levels of PSA between whites and blacks, however, individuals of brown color had a protective factor in relation to whites (PR = 0.26; 95% CI = 0.08–0.82).

In Brazil, several studies have not found differences between black and white individuals [29,30,31,32]. These results can be attributed to race miscegenation in the Brazilian population, but another potential source of bias may include the use of different methodologies to classify individuals into racial groups [29,33,34]. The model of race/ethnicity/racial classification in Brazil is more complex than in other countries such as the USA and European models, the classification is usually self-reported in the ethnic/racial choice by the interviewee or made by the interviewer, using skin color as a basis, among other physical characteristics (facial features, hair texture, lip and nose shape), which can distort the results [33,34]. In the present study, race/ethnicity was assessed by the interviewers for skin color.

Regarding the association of PSA with blood pressure, Loprinzi and Kohli [13] found association only for DBP, when adjusting for age, BMI, race/ethnicity, nicotine concentration, income, marital status, years of study, diabetes, alcohol consumption and use of medicines. Martin et al. [35] report a 5% increased risk for PC for each 11.4 mmHg increase in DBP. Beebe-Dimmer et al. [36] reported a twice as high risk of detecting PC associated with hypertension.

The prevalence of metabolic diseases in our study (22.1%) is higher than that of Dutra et al. [37] (11.7% in older adult men), however it is lower than that of a nationwide study conducted in Brazil [38], and this may have occurred due to the different evaluation methods. Regarding metabolic diseases, diabetes is one of the most prevalent [39,40]. Its association with PSA as a risk factor differs from the study by Gómez-Acebo et al. [3], where they found it as a protective factor for PC in individuals who were not treated (odds ratio = −0.41; 95% CI = 0.22–0.77) or in those whose treatment was with oral antidiabetics (odds ratio = −0.69; 95% CI = 0.52–0.92). This decrease has been linked to the use of anti-diabetic drugs [41], and studies have observed a decrease in the risk of PC among metformin users [42,43].

In our study, a prevalence of 23.2% of alcohol consumption was found, which is in line with that observed in the Brazilian men over the age of 55 years (prevalence of 24.8% of alcohol consumption) [38]. In addition, an association was found between alcohol consumption and high levels of PSA, and this result is consistent with the meta-analysis by Zhao et al. [44] and similar to the study by Gómez-Acebo et al. [3], which carried out a population-based case-control study in Spain with 818 cases of PC and 1006 controls, where men in the second tertile of alcohol consumption (8 and 21 g per day) increased the risk of PC, however, the age of the participants was 20 to 85 years old (odds ratio = 1.46; CI 95% = 1.10–1.93). This result differs from Burton et al. [11], who carried out a cohort study with 1115 men, aged 50 to 69 years, from nine different regions of the United Kingdom with PC, finding no association between alcohol consumption. This divergence can be attributed to the way of assessment or by the age of the participants.

Another factor associated with changes in PSA was the SB (PR = 1.00; 95% CI = 1.00–1.01), which is already associated in the literature as causing several harms [45,46,47,48,49]. Studies have also shown this association between high levels of PSA and the appearance and progression of PC, with a long time in SB [15,50]. Loprinzi and Kohli [13] showed that individuals with high concentrations of PSA were involved in higher CS than those without concentrations of high PSA (635 min/day vs. 611 min/day) and also stated that for every 1 h added SB, the probability of participants having high concentrations of PSA increased by 16%, and for every 1 h added of light physical activity, this risk decreased by 18%. Giovannucci et al. [7] found a strong association between vigorous physical activity and a reduced risk of fatal PC or progress.

The results suggest that certain populations (for example, diabetic or hypertensive), who live alone, consume alcohol and are exposed for a long time to SB, may have higher levels of PSA, and consequently, they are at greater risk of PC. Knowing that SB and hypertension are associated with PC in the literature [35,51,52], SB can increase adiposity and, therefore, increase the risk of developing PC through high levels of sex hormones, insulin resistance and chronic inflammation [53]. High blood pressure can reduce IGF-1 binding protein, which can increase IGF-1 activity and increase the risk of PC [35].

As study limitations, we highlight the lack of information on family history of prostate cancer, and not performing food analysis of participants, as it is a coastal region with high fish consumption, as stated by Pernar et al. [9], and populations with diets rich in fish tend to have a lower incidence of cancer. Another limitation was the non-diagnosis of prostate cancer in the participants, performing only the collection of the PSA level. Finally, the various variables assessed by questionnaires may suffer bias from the instrument and the results are not representative of men in the Brazilian population.

## 5. Conclusions

The study found a high prevalence of high PSA concentrations in the elderly, which can be a worrying factor for their health, since PSA is an important marker for PC. In addition to age, associations of PSA with years of study, race, family arrangement, health status, systolic and diastolic blood pressure, metabolic diseases, alcohol consumption and sedentary behavior were found.

## Figures and Tables

**Table 1 biology-09-00329-t001:** Characteristics of participants.

Variables	Mean (SD)	Minimum	Maximum
Age (years)	70.9 (8.1)	60	93
Body Mass (kg)	72.7 (13.1)	49.7	111.3
Height (m)	1.66 (0.7)	1.50	1.85
Body Mass Index (kg/m^2^)	26.4 (4.4)	18.4	39.5
MVPA (min/day)	515.4 (721.5)	0	3600
Sedentary Behavior (min/day)	447.1 (154.6)	35	960
SBP (mmHg)	144.9 (21.9)	84	221
DBP (mmHg)	79.8 (11.9)	45	109

SD: standard deviation. MVPA: moderate and vigorous physical activity. SBP: systolic blood pressure. DBP: diastolic blood pressure.

**Table 2 biology-09-00329-t002:** Crude and multivariate Prevalence Ratio (PR) for independent variables in relation to the prostate-specific antigen (PSA).

Variables	%	Crude Analysis		Multivariate Analysis **	
		PR [CI 95%]	*p* *	PR [CI 95%]	*p* *
Block 1 Sociodemographic Variables					
Years of Study			0.038		<0.001
Literate	67.4	0.45 [0.22–0.96]		0.23 [0.10–0.95]	
Not Literate	32.6	1		1	
Race/Ethnicity			0.094		0.050
Brown	36.8	0.73 [0.26–2.10]		0.26 [0.08–0.82]	
Black	25.3	1.93 [0.83–4.48]		0.46 [0.18–1.17]	
White	37.9	1		1	
Family Arrangement			0.082		<0.001
Accompanied	83.2	0.50 [0.23–1.10]		0.17 [0.07–0.38]	
Alone	16.8	1		1	
Block 2 Health Variables					
Health Perception			0.047		0.043
Negative	61.1	2.79 [1.01–7.64]		2.45 [1.03–5.82]	
Positive	38.9	1		1	
SBP		1.03 [1.02–1.04]	<0.001	1.02 [1.01–1.04]	0.013
DBP		1.03 [1.01–1.06]	0.008	1.03 [1.00–1.06]	0.042
Metabolic Diseases			0.036		0.044
Yes	22.1	2.20 [1.05–4.59]		2.35 [1.02–5.40]	
No	77.9	1		1	
Block 3 Functional and Behavioral Aspects					
Alcohol Consumption			0.187		0.001
Yes	23.2	1.69 [0.78–3.64]		4.06 [1.77–9.31]	
No	67.8	1		1	
Physical Activity in Transport		0.99 [0.99–1.00]	0.141	0.99 [0.99–1.00]	0.151
Lower Limb Flexibility		1.02 [0.99–1.04]	0.083	1.01 [0.98–1.05]	0.343
Calf Circumference		0.94 [0.86–1.02]	0.148	0.93 [0.85–1.02]	0.138
TMM		0.95 [0.87–1.01]	0.117	0.94 [0.88–1.01]	0.117
Sedentary Behavior		1.00 [1.00–1.01]	0.024	1.00 [1.00–1.01]	0.039

** Adjusted for age and monthly family income. * Wald test. SBP: systolic blood pressure. DBP: diastolic blood pressure. TMM: total muscle mass. CI: confidence interval.

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
