# Peer review of "Prevalence and Factors Associated with High Concentration of Prostate-Specific Antigen: ELSIA Study"

_biology, 2020, doi:10.3390/biology9100329_

Round 1

Reviewer 1 Report

This is a cross-sectional study on the prevalence of high levels of PSA and its association with a number of sociodemographic variables in a population of over 100 men of brazilian origine. The study is well designed and performed and the results are in line with other studies published.

The main concern of this study is about its general interest, since it has been performed on a group of people living in the coast of Brazil and with a fish-based diet. Whether these results are representative of similar populations or of the rest of Brazil men is in question and should be discussed by the authors. It would advise to also include a reference to this fact in the tittle, at least to warn readers that they are dealing with a very specific group of men.

Furthermore, I have also been surprised by results such as the prevalence  of low (or no) alcohol comsumption and the small percentage of patients with metabolic diseases. Are these specific for the population studied? How are these variables in the general brazilian population? Is the questionary to assess metabilic disease validated? This should be also discussed more in deep.  

Author Response

Thank you for the opportunity of submitting the revised version of the manuscript. We are sending a point-by-point answer to the reviewer`s. The changes on the text were made by using the track change mode in MS Word. The manuscript has improved substantially and we believe it is now suitable for publication in the Biology.

Point 1: The main concern of this study is about its general interest, since it has been performed on a group of people living in the coast of Brazil and with a fish-based diet. Whether these results are representative of similar populations or of the rest of Brazil men is in question and should be discussed by the authors.

Response 1: Thank you for your comment. The results are not representative of the Brazilian population. We have included this information on lines 270-271 “and the results are not representative of men in the brazilian population”. Please let us know if this explanation resolves yours doubts in this matter.

Point 2: It would advise to also include a reference to this fact in the tittle, at least to warn readers that they are dealing with a very specific group of men.

Response 2: Thank you for your suggest. We include a reference to regionality in the title. Lines 3-4 “Prevalence and factors associated with high concentration of prostate specific antigen – Study ELSIA”. Please let us know if this explanation resolves yours doubts in this matter.

Point 3: Furthermore, I have also been surprised by results such as the prevalence of low (or no) alcohol comsumption and the small percentage of patients with metabolic diseases. Are these specific for the population studied? How are these variables in the general brazilian population? Is the questionary to assess metabilic disease validated? This should be also discussed more in deep.

Response 3: Thank you for your attention. The results are specific to the population studied. Information on these variables in the Brazilian population was discussed in lines 230-231 “The prevalence of metabolic diseases in our study (22.1%) is higher than that of Dutra et al. [35] (11.7% in older adult men), however it is lower than that of a nationwide study conducted in Brazil [36], this may have occurred due to the different evaluation methods”  and 238-240 “In our study, a prevalence of 23.2% of alcohol consumption was found, which is in line with that observed in the brazilian men over the age of 55 years (prevalence of 24.8% of alcohol consumption) [36]”. Finally, metabolic diseases (yes or no) were assessed by self-report, information on lines 96-97. Please let us know if this explanation resolves yours doubts in this matter.

  1. Dutra, E.S.; De Carvalho, K.M.; Miyazaki, É.; Merchán-Hamann, E.; Ito, M.K. Metabolic syndrome in central Brazil: Prevalence and correlates in the adult population. Diabetol. Metab. Syndr. 2012, 4, 20.

  1. Brasil. Secretaria de Vigilância em Saúde. VIGITEL 2019: Vigilância de Fatores de Risco e Proteção para Doenças Crônicas em Inquérito Telefônico.; Ministério da Saúde: Brasília, 2020.

Reviewer 2 Report

Overview and general recommendation: The present manuscript is about the analysis of the prevalence and the factors associated with high concentrations of the prostate-specific antigen (PSA) in the elderly. This topic is timely, even if not novel. Prostate cancer is age-related cancer, understand the prevalence and factors related to high-level PSA is important to build strategies to decrease PSA, and reduce prostate cancer morbidity and mortality in the elderly. The authors analyzed 96 over than 60 years old male samples and their results showed that the prevalence of high PSA was 21.9% and high levels of PSA was associated with years of study, race/ethnicity and family arrangement, health perception, systolic blood pressure, diastolic blood pressure, metabolic diseases, alcohol consumption, and sedentary behavior. Therefore, elderly prostate cancer patients have a high prevalence of high PSA, and several aspects as above mentioned are associated with high PSA and they are the risk factors of prostate cancer with high PSA. The paper is overall well designed and performed.

Major comments:

The main concern I have about the paper is that the sample size is a little too small. Future studies should increase the sample size.

Minor comments:

There are several abbreviations were not providing the full name at the first appearance.

Author Response

Thank you for the opportunity of submitting the revised version of the manuscript. We are sending a point-by-point answer to the reviewer`s. The changes on the text were made by using the track change mode in MS Word. The manuscript has improved substantially and we believe it is now suitable for publication in the Biology.

Point 1: The main concern I have about the paper is that the sample size is a little too small. Future studies should increase the sample size.

Response 1: Thank you for your comment. For future studies we will try to include all the older adults residents in the municipality.

Point 2: There are several abbreviations were not providing the full name at the first appearance.

Response 2: Thank you for your corrections. We hope to have solved all the problems.